# Fusion of Higher Order Spectra and Texture Extraction Methods for Automated Stroke Severity Classification with MRI Images

**DOI:** 10.3390/ijerph18158059

**Published:** 2021-07-29

**Authors:** Oliver Faust, Joel En Wei Koh, Vicnesh Jahmunah, Sukant Sabut, Edward J. Ciaccio, Arshad Majid, Ali Ali, Gregory Y. H. Lip, U. Rajendra Acharya

**Affiliations:** 1Department of Engineering and Mathematics, Sheffield Hallam University, Sheffield S1 1WB, UK; 2School of Electronics and Computer Engineering, Ngee Ann Polytechnic, Singapore 599489, Singapore; falco_peregrinus14@yahoo.co.uk (J.E.W.K.); e0145834@u.nus.edu (V.J.); aru@np.edu.sg (U.R.A.); 3School of Electronics Engineering, Kalinga Institute of Industrial Technology, Bhubaneswar, Odisha 751024, India; sukanta207@gmail.com; 4Department of Medicine-Cardiology, Columbia University, New York, NY 10027, USA; ciaccio@columbia.edu; 5Sheffield Institute for Translational Neuroscience, University of Sheffield, Sheffield S10 2HQ, UK; arshad.majid@sheffield.ac.uk; 6Sheffield Teaching Hospitals NIHR Biomedical Research Centre, Sheffield S10 2JF, UK; Ali.Ali@sth.nhs.uk; 7Liverpool Centre for Cardiovascular Science, University of Liverpool and Liverpool Heart & Chest Hospital, Liverpool L69 7TX, UK; Gregory.Lip@liverpool.ac.uk; 8Aalborg Thrombosis Research Unit, Department of Clinical Medicine, Aalborg University, 9000 Aalborg, Denmark; 9School of Science and Technology, Singapore University of Social Sciences, 463 Clementi Road, Singapore 599494, Singapore; 10Department of Bioinformatics and Medical Engineering, Asia University, Taichung 41354, Taiwan; 11International Research Organization for Advanced Science and Technology (IROAST), Kumamoto University, Kumamoto 860-8555, Japan

**Keywords:** stroke type classification, Magnetic Resonance Imaging, Support Vector Machine, adaptive symmetric sampling, Higher Order Spectra

## Abstract

This paper presents a scientific foundation for automated stroke severity classification. We have constructed and assessed a system which extracts diagnostically relevant information from Magnetic Resonance Imaging (MRI) images. The design was based on 267 images that show the brain from individual subjects after stroke. They were labeled as either Lacunar Syndrome (LACS), Partial Anterior Circulation Syndrome (PACS), or Total Anterior Circulation Stroke (TACS). The labels indicate different physiological processes which manifest themselves in distinct image texture. The processing system was tasked with extracting texture information that could be used to classify a brain MRI image from a stroke survivor into either LACS, PACS, or TACS. We analyzed 6475 features that were obtained with Gray-Level Run Length Matrix (GLRLM), Higher Order Spectra (HOS), as well as a combination of Discrete Wavelet Transform (DWT) and Gray-Level Co-occurrence Matrix (GLCM) methods. The resulting features were ranked based on the *p*-value extracted with the Analysis Of Variance (ANOVA) algorithm. The ranked features were used to train and test four types of Support Vector Machine (SVM) classification algorithms according to the rules of 10-fold cross-validation. We found that SVM with Radial Basis Function (RBF) kernel achieves: Accuracy (ACC) = 93.62%, Specificity (SPE) = 95.91%, Sensitivity (SEN) = 92.44%, and Dice-score = 0.95. These results indicate that computer aided stroke severity diagnosis support is possible. Such systems might lead to progress in stroke diagnosis by enabling healthcare professionals to improve diagnosis and management of stroke patients with the same resources.

## 1. Introduction

Cerebrovascular accident, commonly known as stroke, is a major cause of death and chronic disability on a global scale [1,2,3]. Cerebral ischemia causes approximately 80% of strokes [4]. Energy depletion and cell death might lead to ischemic brain injuries [5]. These injuries cause diminished functionality of the affected nerve cells, which leads to serious long-term disability. Brain imaging, such as Computed Tomography (CT) and Magnetic Resonance Imaging (MRI), is an important tool for ischemic stroke assessment [6,7]. Trained professionals use brain images to determine the affected lesion and classify the stroke into one of three types: Partial Anterior Circulation Syndrome (PACS), Lacunar Syndrome (LACS), and Total Anterior Circulation Stroke (TACS) [8,9,10]. PACS is a cortical stroke and affects the middle/anterior cerebral artery. A blockage in blood vessels which supply deep brain structures leads to LACS. TACS affects the middle/anterior cerebral areas which are supplied by middle and anterior cerebral arteries [11]. Discriminating between PACS, LACS and TACS yields important information for prognosis and treatment planning [12,13,14,15,16]. Treatment for ischaemic stroke exists in the form of intravenous fibrinolysis, intra-arterial therapies, and mechanical revascularization [17]. Currently, human experts use their knowledge on how the physiology of different stroke types is reflected in image texture to classify stroke severity [10]. Cardiologists use textural information to identify extent, shape, and location of lesions. This information forms the basis for stroke severity classification. Unfortunately, manual image analysis is laborious [10], and prone to inter- as well as intra-operator variability [18,19]. Furthermore, expert analysis is time-consuming [16] which drives up the diagnosis cost. Automated lesion identification and subsequent stroke severity classification can considerably reduce the delineation times and detect lesions accurately [16]. Therefore, the development of computer aided diagnostic systems based on automated brain lesion detection after stroke is an active field of research.

Automated stroke severity classification systems are linked to specific data sources. To differentiate PACS, LACS and TACS in a practical setting, the choice comes down to CT or MRI imaging. Both methods yield a graphical representation of the human brain which contains distinct image objects. Identifying these objects through image segmentation is an important step to extract diagnostically important information. When compared to MRI, CT is more widely used because it is faster and less expensive. However, MRI has a much higher sensitivity for acute ischemic lesions [20]. In addition, MRI examinations can be extended by adding functional information to the anatomical data to form Diffusion-Weighted Imaging (DWI). Diffusion refers to the random Brownian motion of water molecules driven by thermal energy [21]. Acute cerebral ischemic lesions, even in their early stage, have a different water proportion when compared to normal brain tissue. Hence, DWI imaging is more sensitive than other MRI modalities to small water diffusion changes in the acute ischemic brain, especially within 48 hours of the ictus [22,23,24,25,26]. Hence, MRI images are a good data source for automated and semi-automated lesion delineation. Early attempts focused on identifying abnormal voxels in individual T1-weighted MRI scans through Voxel-Based Morphometry (VBM) [27]. Although VBM may be capable of highlighting structural abnormalities that can facilitate lesion delineation by expert raters, it lacks the spatial resolution and statistical power to provide a true replacement for manual lesion delineation [28]. Alternatively, several semi-automated and fully automated procedures for lesion identification have been proposed [28,29,30,31,32,33,34,35,36,37,38,39,40,41]. Rekik et al. have summarized and discussed current research on ischemic stroke lesions in terms of prediction and insights into dynamic evolution simulation models [42]. Lesion identification through segmentation is the first step towards stroke severity classification. However, questions remain as to whether there is sufficient understanding to automate the information extraction processes needed to establish stroke severity. With this question, we shift the focus from fundamental research towards more practical considerations which relate to computer aided stroke severity diagnosis.

In this paper, we test the hypothesis that MRI images of the brain hold computer extractible information to determine ischemic stroke severity. To do so, we extract, quantify, and assess diagnostically relevant information that can assist with automated stroke severity classification. This work is based on the idea that precise identification of structural brain damage in lesion areas is essential to quantify and indeed understand the brain abnormality caused by stroke. The precision requirement led us to choose MRI as the imaging modality. Our work is based on 267 MRI brain images taken from patients who suffered either LACS, PACS, or TACS. Adaptive Synthetic (AdaSyn) was used to balance the class specific feature sets. The information extraction was done with a combination of Discrete Wavelet Transform (DWT) and Gray-Level Co-occurrence Matrix (GLCM) as well as the standalone methods of Gray-Level Run Length Matrix (GLRLM), and Higher Order Spectra (HOS). The analysis methods resulted in 6526 parameters, used as features. The features were assessed in terms of their ability to discriminate LACS, PACS, and TACS. The assessment was done with statistical analysis and classification performance optimization. First, we used the Analysis Of Variance (ANOVA) method to rank the features in terms of their ability to discriminate the disease classes. With these results, the classification process uses 10-fold cross-validation to establish three assessment criteria: (1) classification algorithm, (2) performance, and (3) number of features. As classification algorithm we tested Support Vector Machine (SVM) with different kernel configurations. By averaging the performance across all folds, we found that SVM with Radial Basis Function (RBF) kernel showed the best result: 0.95 Dice score, 93.62% Accuracy (ACC), f1-score of 95.00%. Based on the observation space of 267 brain MRI images, we failed to reject our initial hypothesis. Hence, the results support the idea that automated stroke severity classification is possible. This scientific foundation justifies further investigations on a larger observation space. With such a study we might be able to determine the practical feasibility of automated stroke severity classification by focusing on human factor ergonomics during the diagnostic process. Careful analysis could lead to the understanding that there is scope for a stroke severity classification tool which creates an environment that allows human experts and machine algorithms to work cooperatively on the diagnostic task.

We have structured the remainder of the manuscript as follows. The Section 2 introduces both feature generation and classification methods. The features, extracted from MRI images, were used to train and test the classification methods. Section 3 details the test results. In the Discussion section, we compare our work to previous studies, and we state where we could improve and complement these studies. The conclusion section summarizes our efforts to support the thesis that it is possible to automate the extraction of diagnostically relevant information from MRI images.

## 2. Methods

This section describes the methods used to construct and test a system which extracts diagnostically relevant information from MRI images of the brain. More specifically, we are interested to establish the diagnostic support quality for differentiating LACS, PACS, and TACS. We have addressed this problem with a state-of-the-art design strategy which involved pre-processing, feature extraction, data augmentation, feature ranking, and classification. Figure 1 shows an overview block diagram which indicates how the data flows through the individual processing methods. We have distilled diagnostically relevant information from the pre-processed MRI images with three distinct feature extraction methods. In the first phase GLCM, GLRLM, and HOS methods were used directly on the pre-processed images. In the second phase, the same three methods were used to extract features from different levels of DWT decomposition. After feature ranking, four different implementations of the SVM classification algorithm were used to assess the ability of the features to discriminate LACS, PACS, and TACS. The classifiers differ in terms of their kernel methods; one RBF- and three polynomial-kernels were used. The following sections introduce the processing steps in more detail.

### 2.1. Image Dataset

We have considered 267 slices (PACS 222, LACS 18, and TACS 27) of brain MRI datasets collected from ischemic stroke patients at the Institute of Medical Science and SUM Hospital, Bhubaneswar, Odisha, India. The data was anonymised such that even basic demographics, like age and gender of the patients, are not available. Experts at the hospital selected a single diffusion weighted slice from all the MRI images that were taken during one imaging session from a stroke patient. The selected slices show the stroke lesion and they were used by human experts to support the diagnosis of PACS, LACS, and TACS, on the basis of the clinical features of the patients. The MRI images of the affected brain were acquired with the Signa HDxT 1.5 T Optima Edition machine (GE Healthcare, Waukesha, WI, USA). The first row of images in Figure 2 shows three example MRI images depicting LACS, PACS, and TACS.

### 2.2. Image Resizing and Contrast Limited Adaptive Histogram Equalization

In a first processing step, we resized all 267 MRI images to 200 × 200 pixels. Subsequently, CLAHE was used to increase the difference in luminance which improves visual feature detection. That was necessary because the brain tissue, shown in the MRI images, lacks contrast, which makes it more difficult to distinguish texture morphology. CLAHE is based on histogram equalization, which can help to increase the prominence of textural changes caused by lesions, via contrast enhancement [43]. Adaptive histogram equalization starts with partitioning an image into nonoverlapping 8 × 8 pixel segments known as tiles. Each tile is enhanced so that the resulting histogram matches a uniform distribution. That process is governed by the contrast enhancement limit (0.01), which prevents over-saturation in uniform image regions. Manipulating the tiles independently might introduce artificial boundaries between them. Therefore, in a final processing step, we used bilinear interpolation to smooth the boundaries. The second row of images in Figure 2 shows the example MRI images after CLAHE.

### 2.3. Adaptive Synthetic Sampling

The image acquisition process resulted in an imbalanced dataset where 83.15% of the images came from PACS, 10.11% came from LACS, and only 6.74% came from TACS. The imbalance is a problem for classification algorithms, because it introduces bias towards the larger set [44]. For example, a decision support system can score 83.15% accuracy by classifying all input as PACS. Even without that dump approach, it is difficult for the learning algorithm to focus on the two minority classes, because this conflicts with the generalization requirement. To be specific, an adequate representation of the minority classes results in overtraining the network, which causes a deterioration of the classification performance for unknown data. The ability to infer the correct label from unknown data is impeded when the training process focuses too much on the individual cases.

Obtaining more data from the minority classes would be the best way of solving that problem. Unfortunately, in this case it is not possible because the data was obtained from a medical study, and the results reflect the distribution of stroke severity in patients. Clearly, even when more MRI scans are conducted, it is expected that the distribution of PACS, LACS, and TACS is similar to the original dataset. Hence, the problem must be solved on the data processing level. There are two different approaches to accomplish that. We could introduce a cost factor and make it more ‘expensive’ for the machine learning algorithm to misclassify elements of either LACS or TACS [45,46,47]. This form of intervention requires specific classification algorithms which are aware of the cost associated with specific classes. The second method for reducing the bias is by balancing the dataset through over- and under-sampling [48,49,50]. The cost function requirement limits the choice of classifiers, because not all relevant algorithms come with that functionality. Therefore, we have chosen to resample the dataset with AdaSyn. The method was originally proposed by He et al. [51] and it aims to preserve all of the data while harvesting the benefits of balanced datasets.

### 2.4. Feature Extraction

The DWT [52] was employed on the balanced dataset, wherein the images were decomposed up to 3 levels. Feature extraction was done in 2 phases; (i) before DWT and (ii) after DWT. In (i), the texture features, GLCM [53], GLRLM [53] and HOS [54] were extracted directly from the balanced set of images. The same features were also extracted in (ii).

#### 2.4.1. Discrete Wavelet Transform

DWT was used to decompose the pre-processed images to extract both location and frequency [55,56]. For this work we have used the first three levels of decomposition. Each level of decomposition yields a detailed (D) and an approximated (A) image. The block diagram in Figure 3 shows how an original MRI slice is decomposed to three DWT levels. Hence, after DWT we had seven images for each of the 267 MRI slices: one original and three detailed as well as three approximated images.

#### 2.4.2. Gray-Level Co-Occurrence Matrix

GLCM features establish a relationship between pixels by analyzing images with second-order statistics. During the analysis, the permutation frequency of the pixel brightness is identified [57].

Fourteen linear and nonlinear GLCM features were extracted from the pre-processed MRI images. The linear feature extraction methods were: autocorrelation [57,58], Max Probability [57,59], Dissimilarity [57], Entropy [60], Cluster Shade [57], Sum Average [57], Sum Entropy [57], Sum Variance [57], Difference Variance [57], Difference Entropy [57], Information Correlation Measures 1 and 2 [57].

#### 2.4.3. Gray-Level Run Length Matrix

GLRLM extracts spatial plane features per pixel, relative to the higher-order statistics, wherein a two-dimensional feature matrix is formed at the end [61].

11 GLRLM features, such as the Short Run Emphasis [62], Long Run Emphasis [63], Gray-Level Nonuniformity [63], Run Length Non-uniformity [63], Run Percentage [63], Low Gray-Level Run Emphasis [64], High Gray-Level Run Emphasis [62], Short Run Low Gray-Level Run Emphasis [64], Short Run High Gray-Level Run Emphasis [64], Long Run Low Gray-Level Run Emphasis [62] and Long Run High Gray-Level Run Emphasis were extracted [62].

#### 2.4.4. Higher Order Spectra

HOS based feature extraction methods pick up small structural changes in physiological signals and medical images [65,66,67]. Some of these small structural changes might be due to natural variation, while others might result from pathophysiologic changes caused by a particular disease [68,69,70]. For the current study, we try to classify brain lesions caused by stroke into PACS, LACS and TACS. The classification reflects different pathophysiological processes which result in subtle changes in the MRI image texturing.

HOS feature extraction methods use the Fourier transform of higher order correlations to identify the presence of nonlinear coupling information [71]. These textural features are competent, and hence they were used in our study. To be specific, the following nonlinear features were extracted with HOS methods: entropies 1, 2, 3 as well as HOS phase entropy and HOS mean of magnitude [54]. These features were extracted with phase angels from 1 to 180 in steps of 1°.

### 2.5. Statistical Analysis

Statistical analysis was used for feature ranking, which is a prerequisite for classification-based feature selection. A particular feature can be considered more important if we can rank it among the other features based on some metric. Therefore, a higher-ranked feature is more valuable for classification than a lower-ranked feature. Moreover, ignoring features that have a rank lower than a specific threshold can also increase classification speed.

The Feature extraction yielded 6475 parameters. This number results from 14 GLCM and 11 GLRLM features that were extracted from the seven images (two for each level of DWT decomposition and one original). Overall that resulted in 175 features. In addition, the five HOS features were also extracted from each image. Evaluating these features in one degree phase steps for half a circle resulted in 6300 = 5 × 180 × 7 features. The 6475 computed features were ranked using the F-value obtained from ANOVA [72]. That allowed us to order the feature set from the largest F-value down to the lowest one.

### 2.6. Classification

Training and testing machine learning algorithms involves creating training and testing datasets. That data split might introduce bias and the results are less reliable because not all available data were used for testing. We have used 10-fold cross-validation to reduce bias and to improve test coverage. The method unfolds as follows. Initially, all available data i.e., the feature vectors, are split into 10 parts. From these 10 parts we constructed 10 folds by selecting each of the parts as test data and the remaining nine parts as training data. Sequential forward selection was used on each fold for feature selection [73]. Subsequently, these folds were used to train and test the SVM algorithms. Hence, for each of the four tested classification algorithms we obtained 10 sets of performance measures—one for each fold. Averaging the individual performance measures over the 10 folds serves to reduce bias, and it improves the test coverage [74].

#### 2.6.1. Support Vector Machine Classifier

We have selected SVM algorithms [75] to classify the feature vectors into PACS, LACS, and TACS. A training step was used to construct two hyperplanes in a high-dimensional feature space which separate class-specific training data points [76]. During testing, the hyperplanes were used as decision borders, which facilitated the classification task. In this study, we have used four different methods to construct the hyperplanes. These methods differ from one another in terms of the kernel that was used to map the feature vectors into the high-dimensional space [77]. The first method was based on a first order polynomial kernel, also known as a linear kernel. Two more methods were constructed by increasing the polynomial order to two and three, respectively. The fourth method employs the RBF kernel function which is based on the squared Euclidean distance between the feature vectors.

#### 2.6.2. Performance Measures

At a global level, the classification method is judged successful if it can identify the correct stroke type. This testing results in a 2 by 2 confusion matrix with True Positive (TP), True Negative (TN), False Positive (FP), and False Negative (FN) entries. These entries were used to calculate ACC, Positive Predictive Value (PPV), Sensitivity (SEN), and Specificity (SPE), as defined in the following equations:
(1)$$\begin{array}{cc} \mathrm{ACC} & =(\;\mathrm{TP}\;+\;\mathrm{TN}\;)/(\;\mathrm{TP}\;+\;\mathrm{TN}\;+\;\mathrm{FP}\;+\;\mathrm{FN}\;) \end{array}$$
(2)$$\begin{array}{cc} \mathrm{PPV} & =\mathrm{TP}/(\;\mathrm{TP}\;+\;\mathrm{FP}\;) \end{array}$$
(3)$$\begin{array}{cc} \mathrm{SEN} & =\mathrm{TP}/(\;\mathrm{TP}\;+\;\mathrm{FN}\;) \end{array}$$
(4)$$\begin{array}{cc} \mathrm{SPE} & =\mathrm{TN}/(\;\mathrm{TN}\;+\;\mathrm{FP}\;) \end{array}$$


We also established Dice’s similarity index [78,79] using the following formula:
(5)Dice=2TP/(2TP+FP+FN)


## 3. Results

This section presents the 10-fold cross-validation performance results achieved by the four tested SVM classifiers. These results are instrumental to judge feature quality. As such, feature quality reflects the diagnostically relevant information that can be automatically extracted with signal processing algorithms. Hence, the classification results help us in our quest to determine if, and indeed to what extent, brain MRI imagery contain machine extractable information for stroke severity classification. Table 1, Table 2, Table 3 and Table 4 detail the classification performance for each fold and the average performance, overall folds, of the four tested SVM classifiers.

Table 5 shows the classification results of the SVM classifiers. From the results, it is clear that the SVM RBF classifier obtained the highest classification accuracy of 93.62% and a dice score of 0.95 for the classification of PACS, LACS and TACS images.

## 4. Discussion

Current medical imaging technology allows us to support human expertise during diagnosis and treatment monitoring [80]. With this study, we propose to extend existing technology with decision support algorithms to create a Computer-Aided-Diagnosis (CAD) system. We investigated ischemic stroke severity classification based on standard MRI imagery. Framing the study in such a way allowed us to set the aims and objectives for the proposed decision support algorithms. Within this framework, the main aim of our study is to bring about evolutionary change by providing an adjunct tool for stroke severity classification based on MRI image analysis. To achieve that aim, we must show that it is possible to automate the extraction of diagnostically relevant information from MRI imagery. Therefore, the main objective for this study was to evaluate a range of information extraction methods in terms of their ability to support a stroke severity diagnosis. The evaluation process was structured into statistical and classification performance assessment. The assessment results provide a measure that determines to what extent the main objective was met. Furthermore, these results feed into the wider research community because they provide a way to compare different studies.

Looking back, the initial computer support tools were capable of lesion segmentation in MRI scans [16,81,82]. These methods employed linear decision algorithms, such as thresholding, to accomplish the segmentation task. More recently, these linear methods have been replaced by machine classification methods, because these methods consider nonlinear relationships between information extracted from the MRI images. Studies which investigated lesion segmentation problems show that machine classification outperforms linear decision methodology [15,42,83]. The medical rationale behind these studies was that size, region, and density of the lesion are correlated with the damage caused to brain tissue, which correlates to the cognitive impairments of patients. Addressing this medical need with computer algorithms instead of human expertise, followed the same reasoning as we have put forward for our study. Manual lesion segmentation by a trained professional is referred to as the gold standard against which the automated systems are compared [8,9]. Having automated systems for lesion segmentation offers several advantages. Such a system can be used as an adjunct tool which provides a second opinion on a specific MRI scan. Having such a second opinion, from a tool that does not suffer from inter- and intra-observer variability [18], might lead to a better diagnosis. There is also the economic argument on progress which means to do more with the same resources. For CAD systems, progress implies a shift from human labor to machine work. Lesion segmentation follows a set of rules, and knowing these rules requires theoretical as well as practical training. That training process is a significant cost factor when it comes to health economics. CAD algorithms formalize the lesion segmentation rules such that they steer the execution of a computer program. That computer program needs to be coded only once and it can be deployed multiple times. Usually, this is more cost-effective when compared to training human experts. However, human experts are not limited to analyzing the images; they are also establishing a diagnosis through decision-making. Reaching these decisions requires not only reason but also perception, and indeed, intuition guided by experience. Hence, one way to make further progress is to shift work that requires higher mental capabilities from humans to machines.

Artificial Intelligence (AI) is a technology which can mimic higher mental functions of human experts. On a taxonomy level, machine classification is a subset of AI. However, machine classification methods tend to be referred to as AI when they perform higher mental functions for decision-making. Being capable of reaching that type of decision is distinct from the logical analysis tasks for image segmentation, discussed in the previous paragraph. Combining AI with analysis algorithms creates decision support systems which can execute more tasks on a higher mental level. Inevitably, that kind of progress means to shift more work and indeed responsibility from human experts to machines. Therefore, the performance of such systems is even more critical than it is for analysis support algorithms. When compared to analysis systems, the advice from decision support systems appears later in the diagnosis process, i.e., closer to a decision point; hence there is less room or time for human experts to falsify the machine result. For example, there is a distinct possibility that a decision support system provides the wrong advice, and a human expert will follow that advice, which could lead to negative outcomes for patients. Hence, systems which interpret lesion segmentation results must cope with additional performance requirements. The technology to meet these stricter requirements became available only in recent years; therefore, decision support systems, which provide a statement about the disease itself such as stroke severity classification, were created more recently as compared to analysis systems.

There is a direct relationship between the capability of medical support tools and the responsibility that they incur. Unfortunately, that requirement becomes harder to achieve when medical support tools mimic the higher order mental capabilities of human experts. Therefore, these tools tend to become more complex and less explainable. Looking over the history of MRI image analysis for lesion segmentation, linear image analysis results are straightforward to explain. For example, it might come down to comparing a specific parameter extracted from an MRI image to a specific threshold. From a consideration of nonlinear relationships, machine learning based image segmentation results are less explainable. Supervised machine learning, which was employed by all the reviewed lesion segmentation systems, requires a training step which extracts the parameters for the nonlinear model. Hence, the system performance depends on the training step, and that step might introduce bias and other limitations. When it is difficult to explain the machine results directly, human experts can move into a supervisory position to verify them. Decisions, done as part of an automated lesion segmentation, can be verified relatively quickly through visual inspection. In many cases, seeing the proposed lesion region superimposed over the MRI image is sufficient to accept or reject the segmentation result. Verification of decision support, which involves mimicking higher mental facilities, is more difficult to verify. For example, stroke-type classification to assess stroke severity requires MRI image analysis followed by a diagnostic decision. The verification process is similar to the diagnostic process; hence it takes about the same time. Consequently, there is little or no progress in terms of timesaving. In this case, having an automated stroke severity classification might serve as a second opinion which improves the quality of a diagnostic decision. Real progress is made when human experts can trust automated stroke severity classification systems sufficiently to base their diagnosis on a suggestion from these systems. However, current methods were not designed with sufficient formality to provide the required decision quality, and indeed they were not tested rigorously enough [84]. Simulating lesion fate prediction might help us to understand the physiology of acute stroke, and it could aide in the creation of stroke severity classification systems [85]. We aimed to support the creation of stroke severity classification systems by assessing task-specific feature extraction and AI methods. By doing so, we established that it is possible to automate the extraction of diagnostically relevant information for stroke severity classification from MRI images. This could add to the theoretical foundation for formally developed decision support systems that can bear the responsibility of deciding stroke severity.

Discussing lesion segmentation and automated stroke severity classification from a historical perspective sets the context for reviewing both processing methods and the quality assessment results of relevant research work. We start that discussion with a method developed by Seghier and colleagues [15]. They used probabilistic tissue segmentations obtained from healthy control subjects to estimate mean gray matter and white matter probabilities for each voxel. To identify lesions in patient scans, these estimates were used to compute a metric for quantifying each voxel’s membership to a fuzzy set within each tissue class. A linear threshold method was used to determine the lesion tissue class for an unknown voxel. Wilke and colleagues [16] developed a semi-automated algorithm that uses probabilistic tissue segmentation to construct a set of four feature maps which encode information about tissue composition, tissue homogeneity, shape, and laterality at each voxel. These feature maps were used to construct robust z-score maps, subjected to manual thresholds. A user must combine the results to establish the final lesion delineation. Mitra et al. [86] approached the problem of lesion segmentation with a combination of Bayesian–Markov random fields and random decision forests for voxel-wise classification in multi-spectral MRI volumes. Their system achieved a Dice coefficient of 0.60 ± 0.12. Karthik et al. [87] used transfer learning to segment ischemic lesions. Their deep supervised fully convolutional network achieved a mean segmentation accuracy of 70%. Acharya et al. [88] used HOS entropy features to build a system which is capable of classifying MRI images after stroke into LACS, PACS, and TACS. Their system achieved sensitivity, specificity, accuracy, and positive predictive values equal to 96.4%, 100%, 97.6% and 100%, respectively. Vupputuri et al. [89] investigated the symmetry between left and right brain hemisphere. Their hypothesis was that hemisphere asymmetry provides an important cue for abnormality estimation, and they tested their hypothesis by building an automated lesion segmentation system based on a novel superpixel cluster method. That system achieved a Dice similarity score of 0.704 ± 0.27. Our feature extraction method fuses texture analysis methods with HOS. The resulting features are more expressive when compared to the individual methods.

### Limitations and Future Work

Our current work is an attempt to improve the results of our previous work [88]. We have tested a diverse range of features. However, the best performance was close to the one obtained previously. In the current effort, we have fused HOS based features with texture features extracted through GLCM and GLRLM. The fused feature extraction methods were applied to the image directly, as in our previous work, and the first three levels of DWT decomposition. Our results show that the additional texture feature extraction methods (GLCM and GLRLM) and decomposing an image with DWT fail to improve the classification performance. We have to conclude that HOS based features are most suitable for the classification of brain MRI images, because these methods are able to capture subtle changes in the pixel intensities efficiently. Having realized the discriminative power of HOS for stroke severity classification based on MRI images, there might be scope for investigating higher order cumulants as a feature extraction method. Our current work on HOS can serve as a benchmark with which any future method is compared.

Common to both studies is the fact that we had only 267 MRI images and there were significantly more PACS (222) images than TACS (27), and LACS (18). Having so few images limited our study to information extraction and assessment. We could only show that it is possible to extract information from the MRI images that can be used to differentiate the stroke severity. However, it was not possible to construct a system which can be used in a practical diagnosis support setting, because these images do not reflect depth and breadth of MRI images collected for stroke severity diagnosis. Breadth refers to the fact that every brain and indeed every stroke is different. More images are needed for the machine to learn what constitutes a normal brain and how it is affected by different types of stroke. With the current limited dataset, a system which could extract such deep knowledge would overfit. That means the system would extract relevant and irrelevant information during the training phase and thereby gain no knowledge on how a stroke alters the brain. The failure to gain that knowledge manifests itself in a poor performance when the classification system is tested with unknown MRI images. Significantly more data is needed to force such AI systems to learn only from relevant information which lead to an understanding on how the MRI technology captures the different types of stroke. Another limitation is that we had data from only one center in Bhubaneswar, Odisha, India. Hence, our study might be biased to detecting stroke in a specific ethnic group, because the measurements were carried out in one geographical region. In other words, for a different region, different features might be more relevant. Furthermore, having only 267 MRI images from one center is not deep enough to rule out observer bias. One final thought on the data and its limitations centers on the fact that the MRI slices were carefully selected to show clear-cut textbook examples for the different types of stroke. This limits our study in terms of automatization and generalization. Manual selection of MRI slices introduces inter- and intra-observer variability. Furthermore, it requires expert labor which is a major cost factor. Hence, the proposed automatization is still incomplete, and slice selection is an important topic for future research work. Indeed, looking at the continuum of slices, rather than just one, might reveal more information which can lead to more robust decision support results. The generalization limitation arises also from the expert selection of MRI slices. Having good examples is important for a learning situation because it avoids confusion among the learners. From this perspective, training AI is not different from training human experts. However, for human experts we have ample of evidence that they can extract transferable knowledge from limited and biased data sets, which helps them in a practical setting. There are significantly fewer examples showing that AI has that ability. In that specific sense, we did not investigate whether the proposed feature extraction methods can be transferred to a practical setting.

The classification into PACS, TACS, and LACS helps human experts to reach a diagnosis, which is the cornerstone for managing the disease with treatment and specialized care. In this study we show that AI systems can be used to formalize the identification of different stroke types. However, for future projects we need more data to address the limitations and biases inherent in the current image dataset. Data from more geographical regions are required to address ethnic bias. More data, even from the same center, is required to avoid observer bias and to reflect observable variations adequately. As a result, the AI algorithms for future studies must deal with significantly larger datasets. Experience shows that feature extraction coupled with classical machine learning is not capable of handling the training data needed to prepare AI systems for practical diagnosis support [90]. Therefore, future studies require different AI algorithms, such as Deep Learning (DL). DL is a concept which turns big training data into an advantage for practical decision support systems. This is done by allowing sufficient network depth to capture more relevant knowledge without the need for feature engineering. We suggest that using this type of algorithm will improve the practical relevance for future stroke severity classification systems. However, we are aware that even considering MRI scans from multiple centers of longer time periods might not yield sufficient data to train the most potent deep networks. One way of addressing that problem might be to use transfer learning, which is based on the idea of customizing pre-trained networks [91]. Customizing the network requires less labeled data when compared to training a deep network from scratch. Furthermore, the computational complexity is lower, which makes hyperparameter tuning less time-consuming.

Our final thought on limitations and future work is concerned with the practicalities of having an automated stroke severity classification system in the care pathway. Our initial assumption is that such a system will be useful whenever MRI images are captured to determine the stroke type. In the future, working with multiple centers might help us to determine which stage of the care pathway will benefit most from the application of automated stroke severity classification. This leads to focused research questions, such as: Should automated stroke severity classification be used in acute, sub-acute or chronic settings? Would such a system help in acute treatment with IV or IA thrombolysis or chronic disability management by predicting prognosis or something else?

## 5. Conclusions

With this paper we set-out to answer the question as to whether it is possible to automate ischemic stroke severity classification based on MRI imagery. To investigate that research question, we designed a system which uses HOS, GLRLM, as well as a combination of DWT and GLCM methods for feature extraction. The resulting features were assessed with statistical and classification methods. Statistical assessment allowed us to rank the features. The best features scored a very low *p*-value; therefore, we have to reject the hypothesis that the features were drawn from the same data class. In other words, the statistical analysis indicates that the features reflect different data classes. The classification assessment quantifies the ability of the best features to differentiate between PACS, LACS, and TACS. We found that SVM with RBF kernel achieves: ACC = 93.62%, SPE = 95.91%, SEN = 92.44%, and Dice = 0.95. These performance measures were established based on the rules of 10-fold cross-validation. Hence, the results are statistically robust, and they indicate that fusing texture analysis with HOS produces superior results when compared to individual methods. Furthermore, the established Dice score indicates that our feature extraction method outperforms all MRI based lesion segmentation and stroke severity classification algorithms published in the scientific literature. Therefore, fusing texture and HOS methods can serve as a scientific foundation which indicates that it is possible to differentiate PACS, LACS, and TACS based on MRI images. The proposed feature extraction system helped us to support our hypothesis that it is possible to automate stroke severity classification based on MRI images.

This paper documents a significant step towards automated ischemic stroke severity classification based on MRI imagery. The results provide the necessary confidence to propose further investigations. Data is the most crucial part of these investigations because we need datasets which reflect the diversity encountered during a medical diagnosis. We are confident that the necessary processing tools are available and MRI images contain diagnostically relevant information that can be extracted automatically. Hence, we can shift our research effort from information focused feature engineering to knowledge-based decision support systems. Ultimately, we would like to build a system which understands the MRI image texture for PACS, LACS and TACS. 

## Figures and Tables

**Figure 1 ijerph-18-08059-f001:**
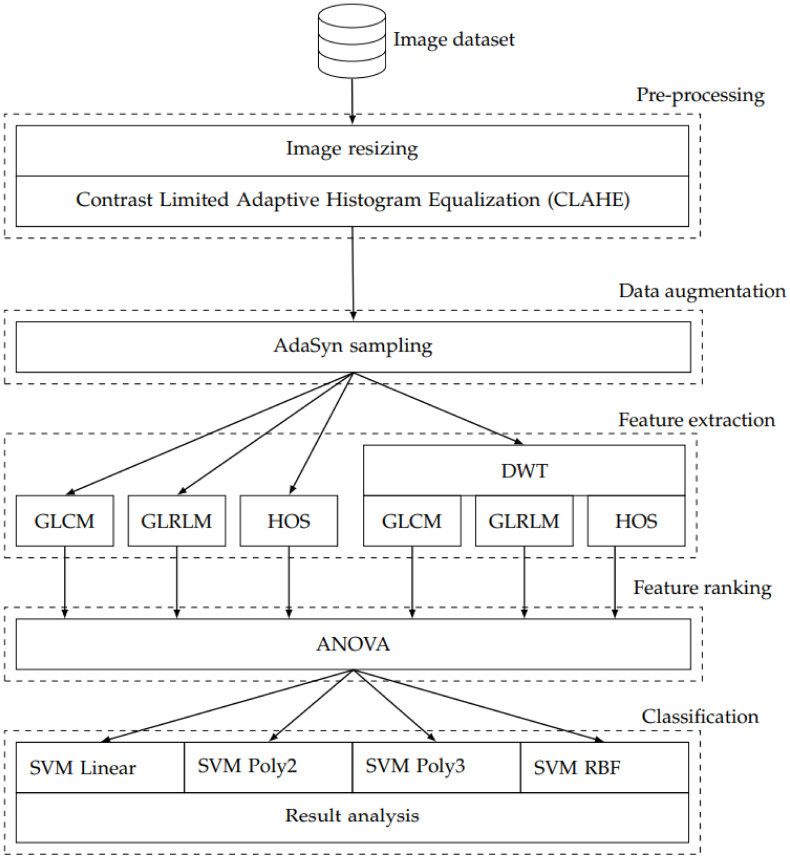
Overview block diagram.

**Figure 2 ijerph-18-08059-f002:**
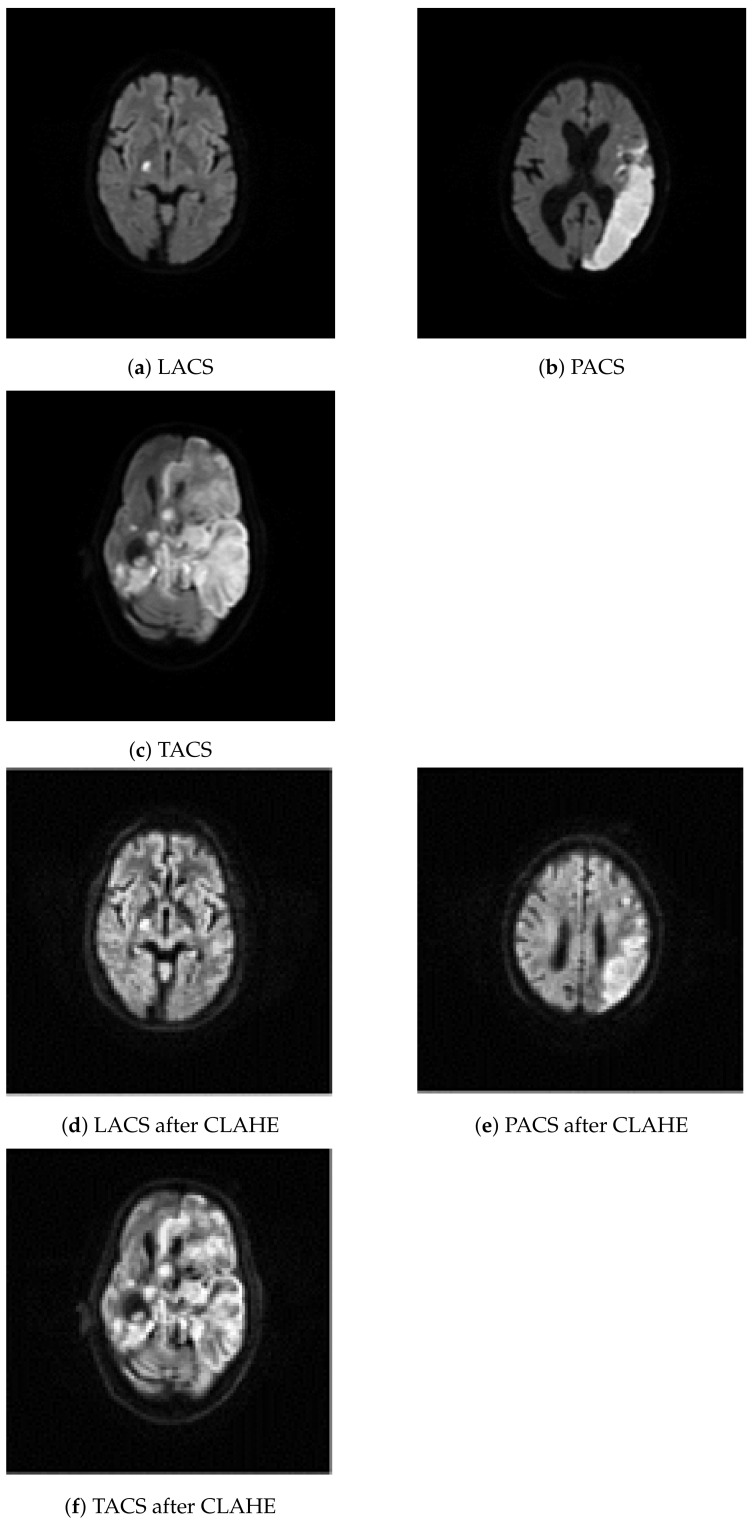
Class specific examples from the image dataset. The original images are shown in the first row: (**a**–**c**). The images were created by processing the original images with contrast adaptive histogram equalization: (**d**–**f**).

**Figure 3 ijerph-18-08059-f003:**
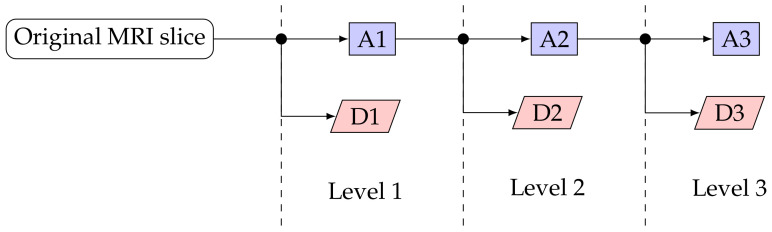
DWT analysis block diagram. DX indicates the detailed image at level X and AX indicates the approximated image at level X.

**Table 1 ijerph-18-08059-t001:** SVM with linear kernel. The first 10 rows in the table document the performance results for each individual fold. The last row states the averaged performance results.

TP	TN	FP	FN	ACC %	PPV %	SEN %	SPE %	Dice
37	21	1	6	89.23	97.37	86.05	95.45	0.91
41	19	4	2	90.91	91.11	95.35	82.61	0.93
38	17	5	6	83.33	88.37	86.36	77.27	0.87
40	21	1	4	92.42	97.56	90.91	95.45	0.94
39	17	5	5	84.85	88.64	88.64	77.27	0.89
39	18	4	5	86.36	90.70	88.64	81.82	0.90
40	19	3	4	89.39	93.02	90.91	86.36	0.92
36	20	2	8	84.85	94.74	81.82	90.91	0.88
38	20	2	6	87.88	95.00	86.36	90.91	0.90
36	21	1	7	87.69	97.30	83.72	95.45	0.90
384	193	28	53	87.69	93.38	87.88	87.35	0.90

**Table 2 ijerph-18-08059-t002:** SVM with a second order polynomial kernel. The first 10 rows in the table document the performance results for each individual fold. The last row states the averaged performance results.

TP	TN	FP	FN	ACC %	PPV %	SEN %	SPE %	Dice
37	22	0	6	90.77	100.00	86.05	100.00	0.93
39	23	0	4	93.94	100.00	90.70	100.00	0.95
40	20	2	4	90.91	95.24	90.91	90.91	0.93
39	22	0	5	92.42	100.00	88.64	100.00	0.94
43	21	1	1	96.97	97.73	97.73	95.45	0.98
40	22	0	4	93.94	100.00	90.91	100.00	0.95
41	21	1	3	93.94	97.62	93.18	95.45	0.95
36	22	0	8	87.88	100.00	81.82	100.00	0.90
40	22	0	4	93.94	100.00	90.91	100.00	0.95
40	22	0	3	95.38	100.00	93.02	100.00	0.96
395	217	4	42	93.01	99.06	90.39	98.18	0.94

**Table 3 ijerph-18-08059-t003:** SVM with a third order polynomial kernel. The first 10 rows in the table document the performance results for each individual fold. The last row states the averaged performance results.

TP	TN	FP	FN	ACC %	PPV %	SEN %	SPE %	Dice
38	22	0	5	92.31	100.00	88.37	100.00	0.94
38	23	0	5	92.42	100.00	88.37	100.00	0.94
40	21	1	4	92.42	97.56	90.91	95.45	0.94
39	22	0	5	92.42	100.00	88.64	100.00	0.94
42	21	1	2	95.45	97.67	95.45	95.45	0.97
41	22	0	3	95.45	100.00	93.18	100.00	0.96
38	20	2	6	87.88	95.00	86.36	90.91	0.90
37	22	0	7	89.39	100.00	84.09	100.00	0.91
37	22	0	7	89.39	100.00	84.09	100.00	0.91
41	22	0	2	96.92	100.00	95.35	100.00	0.98
391	217	4	46	92.41	99.02	89.48	98.18	0.94

**Table 4 ijerph-18-08059-t004:** SVM with RBF kernel. The first 10 rows in the table document the performance results for each individual fold. The last row states the averaged performance results.

TP	TN	FP	FN	ACC %	PPV %	SEN %	SPE %	Dice
38	21	1	5	90.77	97.44	88.37	95.45	0.93
40	23	0	3	95.45	100.00	93.02	100.00	0.96
41	18	4	3	89.39	91.11	93.18	81.82	0.92
40	22	0	4	93.94	100.00	90.91	100.00	0.95
43	20	2	1	95.45	95.56	97.73	90.91	0.97
40	21	1	4	92.42	97.56	90.91	95.45	0.94
42	21	1	2	95.45	97.67	95.45	95.45	0.97
39	22	0	5	92.42	100.00	88.64	100.00	0.94
41	22	0	3	95.45	100.00	93.18	100.00	0.96
40	22	0	3	95.38	100.00	93.02	100.00	0.96
404	212	9	33	93.62	97.93	92.44	95.91	0.95

**Table 5 ijerph-18-08059-t005:** Classification results of the SVM algorithms.

Classifier	Average SEN %	Average SPE %	Average PPV %	Average ACC %	Average Dice
SVM linear	87.88	87.35	93.38	87.69	0.90
**SVM RBF**	**92.44**	**95.91**	**97.93**	**93.62**	**0.95**
SVM polynomial 2	90.39	98.18	99.06	93.01	0.94
SVM polynomial 3	89.48	98.18	99.02	92.41	0.94

## Data Availability

The data presented in this study are available on request from the corresponding author. The data are not publicly available due to ethical restrictions.

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
