# Peer review of "Fusion of Higher Order Spectra and Texture Extraction Methods for Automated Stroke Severity Classification with MRI Images"

_ijerph, 2021, doi:10.3390/ijerph18158059_

Round 1

Reviewer 1 Report

Suggested English grammar/language edits:

Line 24: Remove “might” from “Energy depletion and cell death might lead to ischemic brain injuries”

Line 37: Identify extend, shape. It should be “extent”

Line 198: results are weaker: may prefer results are inconclusive or less reliable.

Suggested Content edits:

Both introduction and Conclusion have a brief description of introduction, method, results and discussion. This leads to repetition of the same information in different sections of the paper. I would request authors to edit the paper in such a way that every subtitle has relevant information to that particular subtitle.

Authors should describe if the MRI helps predicts the severity of stroke in acute, subacute or chronic setting. And how would it help change management of patients. Would it help in acute treatment with IV or IA thrombolysis or chronic disability management by predicting prognosis or something else?

Reviewer 2 Report

Faust et al. investigated an MRI-based automated method to classify stroke severity in a cohort with anterior and small vessel territory ischemic strokes. The study is of interest and I have listed my comments below. As a stroke clinician, I have raised many points about the semantics and clinical relevance of the paper. 

  1. State the number of the stroke subjects included in the study in the abstract section.  
  2. Remove the term “cerebrovascular accident” as it is a commonly misnomer and only creates confusion. 
  3. Replace nerve cells with neurons. 
  4. PACS, LACS, TACS might be what you used in your study but trained professionals also diagnose strokes in posterior circulation and clinically partial and total anterior circulation strokes might not always what we use. So reword the sentence “Trained professionals use brain images to determine the affected lesion and classify the stroke into one of three types: Partial Anterior Circulation Syndrome (PACS), Lacunar Syndrome (LACS), and Total Anterior Circulation Stroke (TACS)”. 
  5. “Treatment for ischaemic stroke exists in the form of intravenous fibrinolysis, intra-arterial therapies, and mechanical revascularization.” We only have thrombolytic therapies and intra-arterial thrombectomy. Also these hyper acute treatments of stroke. So I would suggest to reword the sentence as “Hyperacute treatments of acute ischemic stroke are thrombolytic therapies and intra-arterial thrombectomy.”
  6. “Cardiologists use textural information to identify extend, shape, and location of lesions.” Not cardiologists, neurologists do this. 
  7. “Our work is based on 267 MRI brain images taken from patients who suffered either LACS, PACS, or TACS” I guess you enrolled 267 patients and have only used one MRI from each subject? 
  8. “By averaging the performance across all folds, we found that SVM with Radial Basis Function (RBF) kernel showed the best result: 0.95 Dice score, 93.62% Accuracy (ACC), f1-score of 95.00%. Within the observation space of 267 brain MRI images, we failed to reject our initial hypothesis.” I don’t think results belong in the introduction.
  9. Did the authors seek cross-validation with the gold standard which is the read from radiologist or neurologist?
  10. “Experts at the hospital selected a single diffusion weighted slice from all theMRI images that were taken during one imaging session from a stroke patient. Why only one slice rather than the entire DWI sequence?
  11. What is the cutoff of infarct volume to determine partial vs. total anterior circulation strokes? 
  12. Why posterior circulation strokes are not included?
  13. “The medical rationale behind these studies was that size, region, and density of the lesion are correlated with the damage caused to brain tissue, which correlates to the cognitive impairments of patients.” Not only cognitive impairment, but all kinds of functional impairments such as motor, language, swallowing etc.
  14. “Addressing this medical need with computer algorithms instead of human expertise, followed the same reasoning as we have put forward for our study.” I would suggest to advertise your methods as complimentary to human expertise rather than a replacement. 
  15. The infarct volume is not the only factor that determines the stroke severity. Clinical status of the patient is another crucial factor in determining the stroke severity. This needs to be listed in the limitations. 
  16. “The classification into PACS, TACS, and LACS helps human experts to reach a diagnosis which is the cornerstone for managing the disease with treatment and specialized care.” This classification is actually not used in reaching to diagnosis or determining the management of stroke. 
  17. Another limitation of the study is you haven’t included posterior circulation strokes. This exclusion needs to be discussed and addressed. 
  18. You report results in a clinical cohort. You should include basic demographics such as age, sex, etc. as well as the consent and data collection processes (retrospective vs. prospective etc.). 

Reviewer 3 Report

In this work, the authors present a method for stroke severity classification using texture features from MRI images and employing an SVM classifier.

Unfortunately, I cannot recommend the acceptance of this manuscript in its present context for the following reasons:

  • Although authors claim stroke severity classification (LACS, PACS, TACS), discuss and compare their results in the context of stroke lesion segmentation. This confusion is entirely misleading.

Moreover, they use the dice similarity index to compare their classification accuracy with the segmentation performance of previous efforts in lesion segmentation.

Authors should compare their results with previous works in the same context, that of classification accuracy metrics. Table 6, references  [92],[86] are the only relevant to be compared using the metrics of sensitivity, specificity, and overall accuracy. The rest are irrelevant for comparison since this accuracy (dice similarity index) refers to lesion detection (segmentation accuracy) and not classification (label stroke severity as LACS, PACS, TACS).

Importantly, authors should stress the innovation and contribution of the present work compared to their previous work (ref 86): The same datasets with the same augmentation method and the same classification model have addressed the same classification task with sensitivity, specificity, and accuracy of 96.4%, 100%, and 97.6% respectively.

  • A second important issue is the use of the augmented data in cross-validation. Data augmentation alleviates the training process with unbalance data set. There is no point in doing data augmentation in the test set. Otherwise, the testing accuracy is biased towards high accuracy. The test set should contain only the original cases.
  • Authors extract texture features from the whole brain image without any segmentation process of the brain area. Instead, in the pre-processing, they resize the original MRI image in 200x200 pixels size. However, this is an entirely unnecessary step. One expects a cropping process to remove the extreme image background that is unnecessary for texture feature extraction.
  • Authors claim a huge number of extracted features (6526!). Is this number correct? They should explain how did they arrive with such dimensionality in feature extraction. Also, feature extraction is described differently in different parts of the text:

Lines 106-110: "We have  distilled diagnostically relevant information from the pre-processed MRI images with three distinct  feature extraction methods. The first method involves a combination of DWT and GLCM feature  extraction algorithms. The second and third methods focus on GLRLM and HOS based feature extraction, respectively"

Lines 158-161: "Feature extraction was done in 2 phases; (i) before DWT and (ii) after  DWT. In (i), the texture features, GLCM [53], GLRLM [53] and HOS [54] were extracted directly from  the balanced set of images. The same features were also extracted in (ii)."

  • Authors should give a clear description of their classification task. As it is introduced, one expects a three-class problem i.e LACTS, PACTS, TACS. However, from the presented results (2x2 confusion matrix, sensitivity and specificity), they finally deal with a two-class problem without mention it before in the text. Which is the positive class?

Minor comments

Lines 86-87. "Within the observation  space of 267 brain MRI images, we failed to reject our initial hypothesis..." What do the authors mean by this declaration?

 2.4.1 and 2.4.2:  Interchange the subsections' titles

Round 2

Reviewer 3 Report

The authors have addressed my comments.